# Effects of Hot-Air Drying Temperatures on Quality and Volatile Flavor Components of Cooked Antarctic krill (*Euphausia superba*)

**DOI:** 10.3390/foods14071221

**Published:** 2025-03-31

**Authors:** Ruxin Zhang, Di Yu, Peng Wang, Yujun Liu, Hanfeng Zheng, Lechang Sun, Jie Zheng, Hai Chi

**Affiliations:** 1East China Sea Fisheries Research Institute, Chinese Academy of Fishery Sciences, Shanghai 200090, China; zrx1487466052@163.com (R.Z.); wangp@ecsf.ac.cn (P.W.); zhenghf@ecsf.ac.cn (H.Z.); 2Key Laboratory of Protection and Utilization of Aquatic Germplasm Resource, Liaoning Ocean and Fisheries Science Research Institute, Dalian 116023, China; yudi1986@126.com (D.Y.); lyj7005@126.com (Y.L.); zhengjiessd@163.com (J.Z.); 3College of Ocean Food and Biological Engineering, Jimei University, Xiamen 361021, China; sunlechang@163.com

**Keywords:** Antarctic krill, drying temperatures, quality characteristics, lipid oxidation, volatile flavor components

## Abstract

Hot-air drying is a key step for Antarctic krill (*Euphausia superba*) onboard processing; however, few studies have explored the effects of different drying temperatures on the quality and flavor alternations of Antarctic krill. In this study, we investigated the effects of hot-air drying temperatures on the physicochemical properties and flavor of Antarctic krill. Sensory evaluation, as well as physical and chemical property tests, revealed that Antarctic krill treated with hot-air drying exhibited substantial changes in moisture status, lipid oxidation indices, and *b** value. The sensory evaluation of Antarctic krill under high temperatures (120 °C and 150 °C) showed higher scores (8.45 ± 0.05 and 8.58 ± 0.22, respectively) on smells, whereas the color changes caused by high temperatures also resulted in lower overall sensory evaluation scores. The POV and TBARS values reached the highest at 26.63 ± 0.28 mg/g and 1.45 ± 0.19 mg/100 g, respectively. The *b** value decreased significantly to 22.32 ± 4.56 following 150 °C treatment. Furthermore, a total of 53 volatile compounds were identified by GC-IMS, and the results showed that aldehydes, alcohols, alkanes, ketones, pyrazines, and furans were the main flavor sources of Antarctic krill. At the same time, the GC-MS results showed that the thermal process had no significant effect on the nutrient content of Antarctic krill. The findings obtained in this study provide foundational information for future research on ship-borne processing and high-value utilization of Antarctic krill.

## 1. Introduction

Antarctic krill (*Euphausia superba*) is a shrimp-like zooplankton that is found in the Antarctic Ocean [1,2]. Antarctic krill is high in protein and unsaturated fatty acids and has a massive catch, making it one of the world’s most abundant species in terms of resources [3]. Antarctic krill, therefore, is considered a primary producer in the Antarctic food chain as well as a major source of animal protein owing to its massive fishing capacity and high nutritional value [4,5]. Relative studies have indicated that Antarctic krill compounds have the potential for anti-oxidation, anti-inflammation, anti-obesity, and anti-diabetic benefits on humans [4,5,6,7]. China has been pursuing large-scale exploration and fishing of Antarctic krill since 2009. Currently, Antarctic krill fishery is a significant part of China’s deep-sea and polar fisheries. Given the limitations of global fishery resources, the rational growth and utilization of Antarctic krill have emerged as a critical trend for the long-term development of China’s fishery industry [8].

Cooked Antarctic krill is an essential component in ship-borne processing [9]. A portion of the collected Antarctic krill is frozen and stored onboard as frozen shrimp at low temperatures, whereas the remainder is immediately boiled and processed into Antarctic krill powder, from which Antarctic krill oil is extracted in a land-based factory [10]. Cooked Antarctic krill, the primary output of krill oil and krill powder, requires immediate means for effective preservation and transportation due to limited processing capacity and storage space onboard [9]. One of the most efficient approaches to address these difficulties is to minimize the moisture content of cooked Antarctic krill and to dry it properly.

In general, drying aquatic products ensures their quality by removing excess moisture, efficiently retaining bio-active ingredients, increasing drying yield and meat output, and lowering efficiency and costs [11,12]. Hot-air drying is a common drying technology, but different drying temperatures can affect the rehydration, texture, color, and flavor of the product [13,14,15]. Bai et al. (2022) described the characteristic flavor of Antarctic krill obtained by heat processing, demonstrating alterations in flavor components before and after thermal treatment using E-tongue, E-nose, and gas chromatography-ion mobility spectrometry (GC-IMS) [13]. Sun et al. (2024) recently evaluated the flavor attributes of canned Antarctic krill at various stages and sterilization intensities [16]. Their findings also revealed precise variations in flavor across different Antarctic krill processing methods. However, hot-air drying temperatures effectively impacted the quality, smells, and textures of Antarctic krill, and no systematic investigation until now has been conducted on the impact of different drying temperatures on the quality and flavor alterations of Antarctic krill.

In this study, cooked Antarctic krill were subjected to hot-air drying at various temperatures, and changes in the quality and volatile flavor components of the dried Antarctic krill were analyzed. These findings revealed differences in the quality and volatile flavor components of Antarctic krill associated with various hot-air drying temperatures. This study provides theoretical and scientific information for Antarctic krill development and characteristic flavor evaluation of Antarctic krill products.

## 2. Materials and Methods

### 2.1. Materials

Antarctic krill were collected during the 40th Chinese Antarctic Expedition. Antarctic krill was caught at FAO 48.1 in January and kept at −20 °C. Antarctic krill was delivered to the lab in May and stored at −80 °C for future use.

Normal ketones, including 2-butanone, 2-pentanone, 2-hexanone, 2-heptanone, 2-octanone, and 2-nonanone, were purchased from Alading (Shanghai, China). Nitrogen (99.99% purity) was provided by Haineng Science. Instruments Co. Ltd. (Qingdao, China). Sulfuric acid–methanol solution, thiobarbituric acid, trichloroacetic acid, and sodium sulfate were obtained from Sinopharm Chemical Reagent Co., Ltd. (Shanghai, China). All chemical reagents used in this study were of analytical grade.

### 2.2. Antarctic krill Preparation and Drying Conditions

Antarctic krill were thawed under flowing water, according to the method described by Chi et al. [17]. Selected intact Antarctic krill without black heads, then collected them from boiling water for 15 min before draining at room temperature (25 °C). Approximately 30 ± 2 g of drained Antarctic krill was placed at 80, 100, 120, and 150 °C (designated as 1, 2, 3, and 4, respectively) for hot-air drying. Drying was stopped when the water loss rate (WLR) of Antarctic krill decreased below 0.1%. The drying period was recorded, and the dried Antarctic krill was used for the following tests. Cooked Antarctic krill without hot-air drying treatment was considered as the control group (designated as 0).

### 2.3. Sensory Evaluation

Sensory evaluation was performed by 10 panelists, including 5 women and 5 men, with normal taste sensitivity and professional sensory background following approval from the Ethics Committee of East China Sea Fisheries Research Institute (Approval code: 2023-12-ZX-009). Before receiving written consent, all participants were informed of the research objectives, procedures, and potential risks. The appearance, smell, and texture of cooked Antarctic krill treated at different hot-air drying temperatures were evaluated using the sensory evaluation criteria listed in Appendix A, with the appearance, smell, and texture weighted at 0.5, 0.3, and 0.2, respectively.

### 2.4. Moisture Ratio (MR) and Water Loss Ratio (WLR) Detection

The moisture ratio (MR) and water loss ratio (WLR) of Antarctic krill were determined by using a moisture analyzer (Computrac MAX 4000XL, AMTEK-Brookfield, Middleborough, MA, USA). Briefly, approximately 20 ± 1 g of cooked Antarctic krill was placed on the aluminum pallet of the moisture analyzer. The temperatures were set to 80, 100, 120, and 150 °C, respectively. The MR was determined using Equation (1) [18]. At the same time, the WLR of cooked Antarctic krill was calculated and recorded by tracking changes in the moisture content of cooked Antarctic krill every 16 s. The computation ended when the WLR was less than 0.1%.(1)MR=MrM0
where M_r_ represents the moisture content of real-time drying, and M_0_ represents the initial point of moisture content.

### 2.5. Color Detection

The hot-air-dried Antarctic krill treated at various temperatures was cooled to room temperature and placed flat on a testing plate. The color of dried Antarctic krill was determined using a portable colorimeter (Chroma meter CR400, Konica, Tokyo, Japan), as described by Chi et al. [17].

### 2.6. Low-Field Nuclear Magnetic Resonance (LF-NMR) Measurement

The spin–spin relaxation of dried Antarctic krill was evaluated using an LF-NMR analyzer (MesoQMR23-060H, Shanghai Electronic Technology Co. Ltd., Shanghai, China) with slight modifications, as described by Zhang et al. [19]. Briefly, approximately 3 g of dried Antarctic krill was prepared and placed in 20 mL glass bottles. The parameters were set as follows: TW = 4000 ms, TE = 0.50 ms, NE = 15,000, and NS = 4.

### 2.7. Lipid Oxidation of Dried Antarctic krill

Acid value (AV), peroxide value (POV), and thiobarbituric acid reactive substances (TBARS) were used to evaluate the lipid oxidation of Antarctic krill treated at various hot-air drying temperatures. AV and POV were determined according to the method described by Zhao et al. [20], and TBARS values were measured according to Ghani et al. [21].

### 2.8. Fatty Acids Composition Analysis

The fatty acid composition of Antarctic krill treated at different hot-air drying temperatures was determined using gas chromatography (GC) according to the method described by Nava et al. [22], with slight modifications. Approximately 15 g of dried Antarctic krill was combined with 5 mL of 2% sulfuric acid methanol solution. Following one hour of esterifying the mixture at 70 °C, 750 mL of deionized water and 2 mL of n-hexane were added to the mixture. The mixture was magnetically stirred for 1 min before being placed in a cube containing 2 g of sodium sulfate. The mixture was maintained at room temperature until a layer was formed. Finally, 2 mL of the sample separated from the n-hexane layer was analyzed for fatty acid composition using GC. C19:0 (nonadecanoic acid), at a concentration of 10 μg/μL, was used as the internal standard. The fatty acid composition was analyzed according to Equation (2).(2)$$Cx\;\left(g/100g\right)=\frac{C_{c19}\times S_{x}}{S_{c19}}$$
where *C_x_* represents the concentration of fatty acid of the tested sample, *S_x_* represents the peak areas of the tested sample, *C_c_*_19_ represents the concentration of nonadecanoic acid, and e *S_c_*_19_ means the peak areas of nonadecanoic acid.

### 2.9. Gas Chromatography-Ino Mobility Spectrometry (GC-IMS) Analysis

The volatile flavor compounds in dried Antarctic krill were analyzed using GC-IMS (FlavourSpec, Gesellschaft für Analytische Sensorsysteme mbH, Dortmund, Germany), as described by Miao et al. [23] and Jiang et al. [24], with slight modifications. Dried Antarctic krill (about 3 ± 0.2 g) was placed in a 20 mL headspace vial and incubated at 60 °C for 20 min with a rotation speed of 500 rpm. The injection volume was 500 μL, and the injection needle temperature was set to 85 °C. The GC settings were configured as follows: the column temperature was 60 °C, the carrier gas was high-purity nitrogen (purity ≥ 99.999%), and the program pressure increase, the initial flow rate was 2 mL/min, maintained for 2 min, linearly increased to 10 mL/min within 8 min, linearly increased to 100 mL/min within 10 min, and linearly increased to 150 mL/min within 10 min. Chromatography run time: 30 min; injection port temperature: 80 °C.

### 2.10. Gas Chromatography-Mass Spectrometry (GC-MS) Analysis

The volatile flavor compounds in dried Antarctic krill were quantified using GC-MS (Agilent Technologies, Santa Clara, CA, USA) with a polar capillary column (HP-5MS, 30 m × 0.25 mm × 0.25 μm) according to a previously described method with some modifications [25]. Dried Antarctic krill (2 ± 0.2 g) was placed in a 20 mL headspace vial and incubated at 60 °C for 50 min, followed by extraction for 40 min. The heating program was initially maintained at 50 °C for 3 min, then increased to 250 °C at a rate of 5 °C/min and maintained for 10 min. The collection range was 40–400 m/z. Cyclohexanone (20 μL, 100 μg/mL) was used as the internal standard.

### 2.11. Statistical Analysis

At least three replicates of parallel trials were conducted for each experiment. All data are expressed as mean values ± standard deviations, and one-way analysis of variance (ANOVA) was performed using SPSS (version 26.0; SPSS Inc., Chicago, IL, USA), with *p* < 0.05 considered as statistically significant. The GC-IMS data were evaluated using the VOCal software (VOCal 0.1.3), which includes the GC retention index data (NIST 2020), IMS migration time database retrieval and comparison, and qualitative analysis. The Reporter, Gallery Plot, and Dynamic PCA in VOCal data processing software were used to generate three-dimensional spectra, two-dimensional spectra, differential spectra, fingerprint spectra, and PCA spectra of volatile components for comparing volatile organic compounds between groups.

## 3. Results

### 3.1. Sensory Evaluation of Cooked Antarctic krill Treated at Various Hot-Air Drying Temperatures

The sensory evaluation scores of cooked Antarctic krill treated at various hot-air drying temperatures are presented in Table 1. The results showed that varying the hot-air drying temperature had a significant effect on the sensory scores of cooked Antarctic krill. High-temperature hot-air treatment significantly altered the flavor of Antarctic krill. Group 4 had the highest score (8.58 ± 0.22) for smells, followed by groups 3, 2, 1, and 0 (8.45 ± 0.05, 7.80 ± 0.25, 7.05 ± 0.39, and 3.53 ± 0.47, respectively). Furthermore, higher temperatures negatively affect the appearance and texture of cooked Antarctic krill, resulting in the lowest sensory scores (5.62 ± 0.28 and 6.44 ± 0.10 for groups 3 and 4, respectively). Due to the high weighting of appearance indicators, groups 1 and 2 exhibited the highest sensory scores for Antarctic krill (8.09 ± 0.09 and 8.00 ± 0.29, respectively).

### 3.2. Moisture Content (WC) and Water Loss Ratio (WLR) Analysis

Drying kinetics provide valuable insights into the drying behavior under various conditions, such as temperature and vacuum pressure [26]. Figure 1 illustrates the drying properties of cooked Antarctic krill treated at various hot-air-drying temperatures. The WLR of cooked Antarctic krill rapidly altered under the high hot-air drying temperature (group 4), reaching a maximum value of 8.15 ± 0.02% in a short time, and then rapidly decreased until the completion of the experiment (Figure 1a). At low temperatures (group 1), the WLR was 1.58 ± 0.00%, indicating a modest trend. Notably, the WLR of cooked Antarctic krill in group 2 before 304 s was higher than that of group 3. Antarctic krill exhibited the highest WLR (5.92 ± 0.01%) in group 2 and 3.37 ± 0.01% in group 3, respectively.

The drying temperatures for groups 1-4 resulted in termination durations of 6083.00 ± 3.00, 3764.00 ± 4.00, 3720.00 ± 8.00, and 2100.00 ± 4.00 s, respectively (Appendix A). Accordingly, Figure 1b shows that the WC of Antarctic krill decreased the fastest at high temperatures (group 4) and slowly at low temperatures (group 1). Before 304 s, the WC of cooked Antarctic krill was higher in group 2 than group 3. This finding was also consistent with the WLR of Antarctic krill. The WC of the four types of cooked Antarctic krill (80, 100, 120, and 150 °C) at various drying temperatures reached 26.45 ± 0.03%, 26.15 ± 0.02%, 24.20 ± 0.01%, and 22.71 ± 0.01%, respectively, at the termination point.

### 3.3. Water Distribution

Water migration and distribution in cooked Antarctic krill were analyzed using LF-NMR (Figure 2). As shown in Figure 2, all five groups consisted of bound water (T_21_), immobile water (T_22_), and free water (T_23_). The distribution of these three water components in the control group was more obvious based on the proportion of peaks, indicating that T_21_, T_22_, and T_23_ appear between 0.1–1 ms, 1–100 ms, and 100–1000 ms, respectively.

After hot-air treatment, the bound water and free water contents in cooked Antarctic krill were significantly reduced, and the immobile water content increased. This indicates that hot-air drying changes the binding state of water molecules in Antarctic krill, thereby causing water migration. First, as the drying temperature increased, the binding state of bound water was destroyed, and most Antarctic krill cells were no longer intact, causing the water molecules bound to the polysaccharides on the cell surface to dissociate and the T_21_ peak to vanish. At the same time, free water molecules in Antarctic krill obtained sufficient thermal energy and boosted their activity, and the free water between the krill surfaces transformed from liquid to gas and evaporated, resulting in the disappearance of the T_23_ peak. On the other hand, the T_22_ peak was divided into two: one peak shifted toward a larger relaxation time because the stability of the binding between some immobile water molecules in the cell and macromolecules decreased after heating, and the water molecules transformed from immobile water to free water. In addition, the second batch of samples did not split into peaks, indicating that no new type of immobile water was formed at this temperature, which may explain why the WC and WLR of Antarctic krill at 100 °C were higher than those at 120 °C.

### 3.4. Lipid Oxidation Analysis

Figure 3 displays the lipid oxidation values of cooked Antarctic krill after hot air drying. The results revealed no significant differences in AV among the five groups. The AV values for the five groups were 7.31 ± 0.28, 7.39 ± 0.19, 7.30 ± 0.38, 7.52 ± 0.08, and 7.78 ± 0.12 mg/g, respectively. There were no significant differences in the POV values between the control group and group 1 (*p* > 0.05). The POV values increased as the temperature increased, with group 4 having the highest value (26.63 ± 0.28 mg/g). This indicates that temperature has a significant effect on the POV of cooked Antarctic krill. Furthermore, hot-air-dried cooked Antarctic krill had higher TBARS levels than the control group, but there was no significant difference in TBARS values between group 2 and the control group. Group 1 exhibited the highest TBARS value at low-temperature settings (1.45 ± 0.19 mg/g), whereas groups 3 and 4 had values of 1.41 ± 0.17 and 1.41 ± 0.41 mg/g, respectively.

### 3.5. Color Changes

Table 2 depicts the color variations observed in cooked Antarctic krill treated at various hot-air drying temperatures. The results revealed no significant differences in *L** (lightness) among the five groups. Similarly, with the exception of group 4, there was no significant difference in *b** (yellowness or blueness) findings among the other groups. Group 4 exhibited a lower *b** (22.32 ± 4.56) than the other four groups. The *a** value (redness or greenness) shifted as the drying temperatures increased. The color difference of cooked Antarctic krill was not significantly affected by hot-air drying temperatures of 80–120 °C, as evidenced by the fact that the change in *a** values was not different between groups 2 and 3 or the control group, and there was no significant difference between groups 1 and 4. The color difference was considerably altered by increasing the temperature to 150 °C, though, as both *a** (14.15 ± 2.51) and *b** (22.32 ± 4.56) displayed a declining trend.

### 3.6. GC-MS Analysis

Table 3 shows the quantitative analysis of the volatile flavor components in cooked Antarctic krill treated at various hot-air drying temperatures. Approximately 46 volatile flavor compounds were discovered, including aldehydes, alcohols, alkanes, ketones, pyrazines, and furans. The highest n-hexane content was found in each group (25.46 ± 1.47, 31.75 ± 4.57, 27.50 ± 4.75, 40.70 ± 7.06, and 45.96 ± 6.61 mg/kg), followed by furan tetrahydro-3-methyl (3-MTHF) and furan tetrahydro-2-(methyl) (2-MTHF). In comparison to group 1, the n-hexane value increased as the hot-air drying temperature increased. In contrast, as the hot-air drying temperature decreased, the quantity of furan and tetrahydro-3-methyl increased. Except for group 4, there was no significant difference in the concentrations of methylamine, N,N-dimethyl (DMA), with a value of 7.31 ± 3.76 mg/kg, suggesting that increased hot-air drying conditions (150 °C) may affect Antarctic krill quality. Although the values of other volatile flavor compounds were less than 1 mg/kg, the total amount of these compounds showed a clear increasing trend with the increase in drying temperature. Among them, ketones such as 2 (5H)-furanone, 5-ethyl- and cyclohexanone, 4-methyl-, and alcohols such as 1-pentanol and 1-penten-3-ol, 4-methyl-, under high-temperature hot-air drying temperatures were considerably different from the other groups (Appendix A).

### 3.7. GC-IMS Analysis

The five samples yielded 83 peaks, including 53 qualitatively identified volatile flavor components, such as monodimers and dimers (Figure 4a). There were 18 aldehydes, 10 alcohols, 3 esters, 2 furans, 2 terpenes, 10 ketones, 4 acids, 2 pyrazines, 1 alkane, and 1 sulfide-containing molecule. Furthermore, 14 compounds remained unidentified. The differences in volatile flavor components between groups 1 and 2 were not significant among the five groups, whereas the differences in volatile flavor compounds between group 3 were significant. The GC-IMS data, which included volatile flavor compounds, were analyzed using PCA to highlight the differences between cooked Antarctic krill treated at various hot-air drying temperatures (Figure 4b). As shown in Figure 4b, groups 0, 3, and 4 were well separated, with significant differences across the groups. The variance contributions of PC1 and PC2 were 51% and 23%, respectively. The volatile flavor compounds with the highest concentrations in each sample were clustered together, as shown in Figure 4c. The volatile flavor compounds in area A concentrated in group 3 were 3-methyl-2-butenal, E-2-pentenal, hexanal, 2-methyl-2-propanal, 2-hexenal, and heptanal. The volatile flavor compounds in area B of group 4 also had the highest concentrations, mostly 2-methylbutanal, 3-methylbutanal, 2-heptanone, 2-hexanone, and propanol.

### 3.8. Fatty Acids Profiles

The fatty acid profiles of cooked Antarctic krill treated at various hot-air drying temperatures are listed in Table 4. A total of 22 fatty acids were measured, including 12 saturated, 5 monounsaturated, and 5 polyunsaturated fatty acids. Except for mystic acid (C14:0), the study found no significant difference in the amounts of 12 other types of saturated fatty acids among the five groups. Among the five types of monounsaturated fatty acids, palmitoleic acid (C16:1) and cetoleic acid (C22:1n9) exhibited different values compared to those of the control group, indicating that the hot-air drying temperature affected the contents of the two monounsaturated fatty acids. Furthermore, the values of docosahexaenoic acid (C22:6n3, DHA) in group 2 differed from those of the other groups, but there was no difference in the values of other polyunsaturated fatty acids between the control and experimental groups, indicating that different hot-air drying temperatures had no significant effect on the polyunsaturated fatty acids of cooked Antarctic krill.

## 4. Discussion

Antarctic krill is recognized as the world’s largest source of animal protein owing to its enormous mass and nutritional value [27]. China has harvested and developed Antarctic krill for decades. Given the considerable distance between China and the Antarctic Ocean, as well as the unique autolysis features of Antarctic krill, Antarctic krill must be treated onboard to ensure its quality to the greatest extent possible [9,28]. Antarctic krill normally undergo processing via two procedures: heating and drying. The thermal approach is primarily used to inactivate autolytic enzymes of Antarctic krill, followed by drying, which causes water loss. The final products of Antarctic krill contain low moisture levels (<30%), high lipid contents, and distinct flavors.

Hot-air drying of Antarctic krill is a widely used technology and an essential stage in ship-borne processing [29,30]. In this study, dried Antarctic krill were produced using different hot-air drying temperatures, and dried Antarctic krill with less than 30% moisture content was obtained by varying the treatment time. To date, the moisture content of dry-cooked Antarctic krill has been irregular. Based on the moisture requirements of dried precooked shrimp in China, high-quality dried precooked shrimp have an average moisture content of 23-35% [5]. Our investigation found that cooked Antarctic krill produced WC values of 26.45 ± 0.03%, 26.15 ± 0.02%, 24.20 ± 0.01%, and 22.71 ± 0.01%. This water content could efficiently inhibit the formation of bacteria in precooked shrimp, thereby extending its shelf life and ensuring its quality [31,32]. The number of molds in Antarctic krill treated with different hot-air drying methods should be analyzed in future work. Consequently, the drying periods were approximately 6083, 3764, 3720, and 2100 s, respectively. Zheng et al. (2024) employed a two-step process to determine the drying characteristics of Antarctic krill meal. Lower drying temperatures (75 °C and 95 °C) and vacuum pressures (20, 60, and 101 kPa) were used. They reported that WC remained less than 10% after drying and that the treatments took between 15,600 and 22,800 s to complete [9]. These drying periods were longer than those observed in the 80 °C and 100 °C treatment groups in our study. These differences may be attributed to the fact that lower drying temperatures cannot rapidly modify the water binding pattern in Antarctic krill, resulting in more free water being unable to quickly turn into immobile water and then volatilize and lose. Each drying procedure should be analyzed and compared in terms of quality changes and moisture status in future studies.

Working onboard normally has two characteristics: minimal energy use and high production outcomes. It is critical to strike a balance between the drying temperature, which is essential for minimizing the production space onboard, and the preservation of volatile flavor components, which contribute to sensory properties. Therefore, sensory evaluation is frequently used to assess the quality of Antarctic krill aboard. In our study, the color and texture of Antarctic krill treated at high temperatures (120 °C and 150 °C) remained poor, with low sensory scores. This finding is attributed to the small individual size of the Antarctic krill [8]. However, the flavor of Antarctic krill under high-temperature conditions also has a strong sensory impact. This could be because the flavor compounds that are created when the saturated fatty acids in Antarctic krill oxidize primarily consist of volatile compounds, such as furans, acids, ketones, aldehydes, and esters. These flavors, which have a rich, meaty, and animal-like scent, are produced by processes such as the Maillard reaction and follow a free-radical reaction pathway during fat oxidation. Consequently, taste parameters should be introduced to evaluate the sensory perception of Antarctic krill in the future.

Hot-air drying also has an impact on the volatile flavor components of Antarctic krill. In this study, the aldehyde and ketone contents of cooked Antarctic krill treated at higher drying temperatures (groups 3 and 4) differed considerably from those of the other groups. This difference might be attributed to the oxidation of fatty acids and the generation of peroxides. These peroxides are then degraded into smaller molecules such as aldehydes and ketones. Chen et al. revealed that saturated fatty acids, such as C6-C10, can emit fishy and other flavors at high concentrations [33]. Our investigation detected 18 aldehydes, which was higher than that reported by Jiang et al. for Antarctic krill paste [24]. These results may be related to the oxidation of unsaturated fatty acids and the production of additional aldehydes in Antarctic krill at high temperatures. As a result, future studies should focus on the association between the degree of oxidation of unsaturated fatty acids and the volatile flavor components found in Antarctic krill. However, higher hot-air drying temperatures result in the introduction of undesirable flavor compounds. In the present study, a higher DMA value was observed. This chemical is an irritating gas with a fish oil odor that is commonly found in aquatic products [34]. Oxidation products are useful indicators of the quality of aquatic products [35,36]. Therefore, proper DMA oxidation management is an effective method to maintain Antarctic krill quality.

The flavor of dried Antarctic krill is primarily composed of aldehydes and ketones, with sulfur-containing and heterocyclic chemicals that supplement the mix. Aldehydes are produced mostly by the oxidation and breakdown of polyunsaturated fatty acids (such as EPA and DHA) and have a grassy, greasy, or fishy flavor. Low-threshold flavor compounds have a considerable effect on the overall flavor. Ketones are mostly derived from lipid beta-oxidation or Maillard reaction byproducts and have a creamy, fruity, or moldy scent. In our investigation, aldehydes such as 2-butenal, 2-pentenal, 2-hexenal, and ketones such as 3-pentanone and cyclohexanone all increased significantly, showing that they most likely contributed to the flavor of Antarctic krill after thermal processing. Similar results were observed for Antarctic krill powder and paste [9,24].

Antarctic krill is rich in lipids, thus, it oxidizes and degrades during drying [37]. Several studies have shown that lipid oxidation in Antarctic krill occurs via various thermal processes. In this study, we first evaluated the changes in AV, POV, and TBARS in Antarctic krill treated at various hot-air drying procedures. The results indicated increasing trends in POV, which is in agreement with previous findings [38,39,40]. POV increased the most considerably of all indicators, which might be attributed to the drying time and temperature. Xu et al. confirmed a positive correlation between increased POV values in Antarctic krill and drying time [41]. Zhao et al. employed several drying processes to investigate lipid oxidation in Antarctic krill [20]. The results revealed that the POV values decreased in the early phases of the low-temperature drying treatment and increased as storage duration. Furthermore, we investigated variations in fatty acid content in Antarctic krill. Our findings revealed that C14:0, C16:0, and C18:1 were the most saturated and monounsaturated fatty acids, respectively. Antarctic krill contained high quantities of C20:5n3 (EPA) and C22:6n3 (DHA). There was no significant difference in EPA values between the control and experimental groups. However, the amount of EPA + DHA in Antarctic krill increased as the drying temperature increased, with group 3 having the highest content at 4.91 g/100 g. Liu et al. evaluated the fatty acid composition of Antarctic krill meal [30]. They found that boiling approaches increased the overall quantity of EPA + DHA in free fatty acids by up to 3.69%, whereas drying caused a modest decrease in EPA + DHA values, indicating that EPA and DHA were oxidized in the lipids during drying. A previous study by Chi et al. demonstrated that lipid oxidation indicators can be utilized to predict the shelf life of Antarctic krill products [17]. As a result, future studies will focus on the lipid oxidation process and shelf life of cooked Antarctic krill treated at different hot-air drying temperatures using lipid oxidation indicators.

Drying temperature is a core parameter for Antarctic krill processing on board, which requires optimization based on utilization, equipment conditions, and environmental sustainability. Future studies can concentrate on the use of intelligent temperature control systems and multi-stage drying procedures to increase the commercial and ecological value of Antarctic krill resources. Therefore, the findings in this study serve as a platform for future research on ship-borne processing and high-value usage of Antarctic krill.

## 5. Conclusions

In this study, different hot-air drying temperatures (80, 100, 120, and 150 °C) were shown to influence both the qualitative characteristics and volatile flavor components of cooked Antarctic krill. Compared to the other groups, group 1 had the highest sensory scores and longest drying time. Simultaneously, heating process significantly changed the water content and state of Antarctic krill. Under high-temperature conditions, the lipid oxidation indexes and *b** value of cooked Antarctic krill were significantly altered, with POV increasing and *b** decreasing as temperature increased. In addition, 53 volatile flavor compounds were identified by GC-IMS, aldehydes, and ketones contributed to a significant proportion of the discovered volatile flavor compounds, with n-hexane comprising the largest amount of each component, showing that varying hot-air drying temperatures can affect quality attributes and flavors. Lower hot-air drying temperatures (groups 1 and 2) may improve the sensory quality of cooked Antarctic krill. However, longer drying times may result in fewer distinct flavors. In contrast, higher hot-air drying temperatures (groups 3 and 4) increased the volatile flavor compounds of Antarctic krill while shortening the drying time, but sensory scores and fatty acid oxidation indexes declined dramatically. Considering the energy consumption of ships and the quality requirements of Antarctic krill, cooked Antarctic krill can be dried onboard by hot-air drying at temperatures of 100 or 120 °C.

## Figures and Tables

**Figure 1 foods-14-01221-f001:**
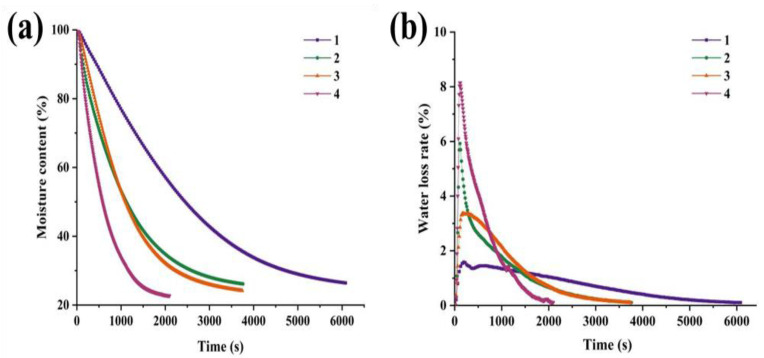
Drying characteristics of cooked Antarctic krill treated at various hot-air drying temperatures. (**a**) moisture content, and (**b**) water loss rate. 1, 2, 3, and 4 present different hot-air drying temperatures at 80, 100, 120, and 150 °C, respectively.

**Figure 2 foods-14-01221-f002:**
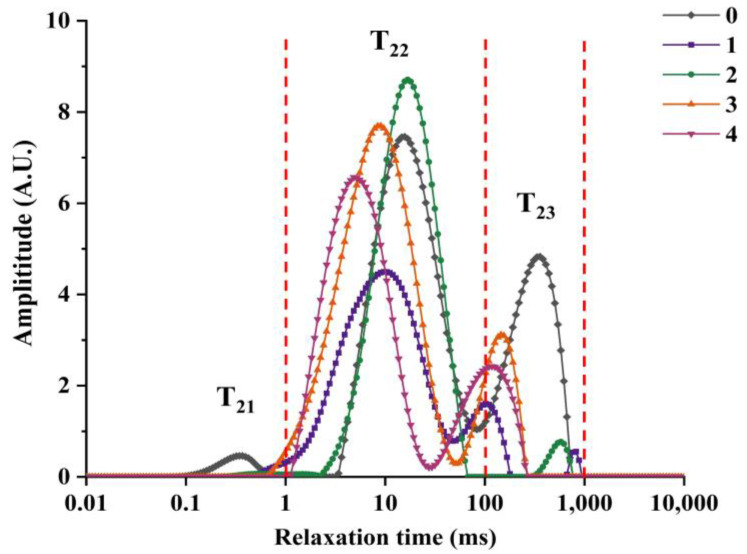
Water distribution of cooked Antarctic krill treated at different hot-air drying temperatures. 1, 2, 3, and 4 present different hot-air drying temperatures at 80, 100, 120, and 150 °C, respectively; 0 presents an untreated sample as a control.

**Figure 3 foods-14-01221-f003:**
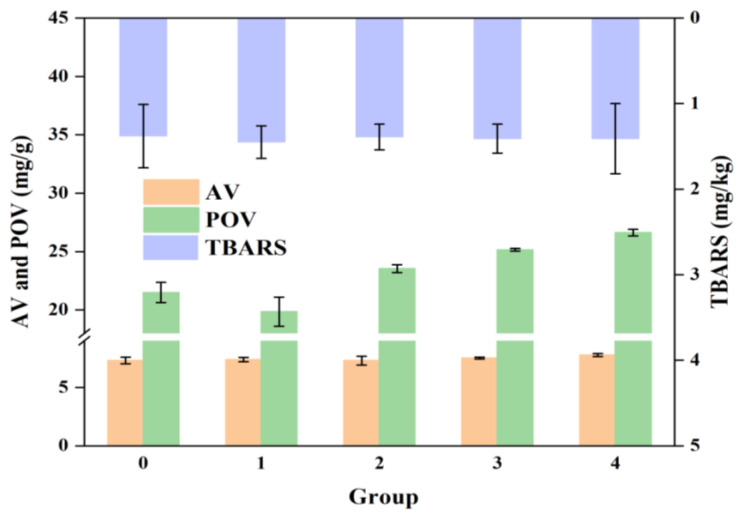
Lipid oxidation of cooked Antarctic krill treated at different hot-air drying temperatures. 1, 2, 3, and 4 present different hot-air drying temperatures at 80, 100, 120, and 150 °C, respectively. 0 presents an untreated sample as a control.

**Figure 4 foods-14-01221-f004:**
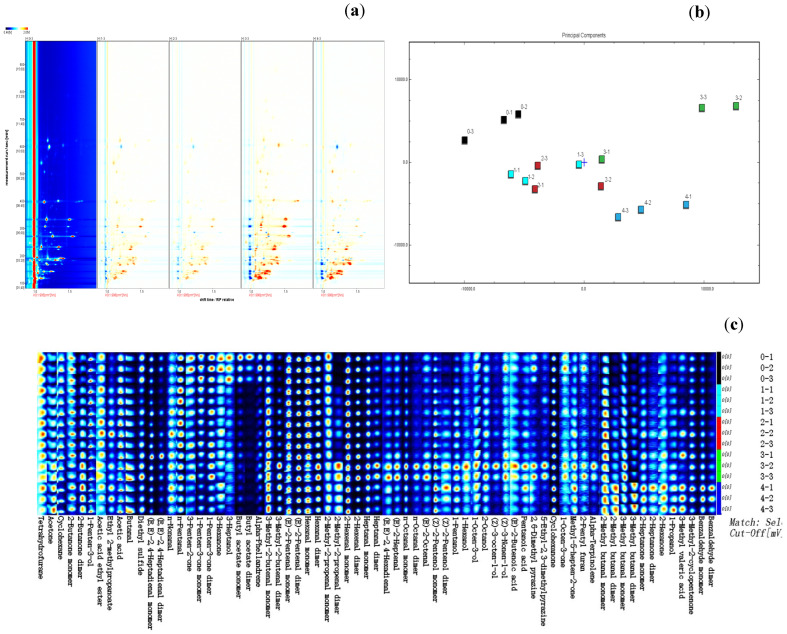
Characteristics of volatile flavor compounds in different cooked Antarctic krill treated at various hot-air drying temperatures identified by GC-IMS. (**a**) comparison of differences, (**b**) PCA chart, and (**c**) fingerprint. 1, 2, 3, and 4 present different hot-air drying temperatures at 80, 100, 120, and 150 °C, respectively. 0 presents an untreated sample as a control.

**Table 1 foods-14-01221-t001:** Sensory evaluation scores of cooked Antarctic krill treated at different hot-air drying temperatures.

Groups	Appearance	Smells	Texture	Total Scores
0	8.50 ± 0.05a	3.53 ± 0.47d	8.05 ± 0.17a	6.94 ± 0.16b
1	8.73 ± 0.08a	7.05 ± 0.39c	8.05 ± 0.11a	8.09 ± 0.09a
2	8.00 ± 0.41b	7.80 ± 0.25b	8.28 ± 0.13a	8.00 ± 0.29a
3	4.45 ± 0.15c	8.45 ± 0.05a	8.40 ± 0.14a	6.44 ± 0.10c
4	3.93 ± 0.23d	8.58 ± 0.22a	5.40 ± 0.74b	5.62 ± 0.28d

Note: 1, 2, 3, and 4 present different hot-air drying temperatures at 80, 100, 120, and 150 °C, respectively. 0 means the control group that samples untreated by hot-air drying. Significant variations between samples are indicated by different letters (*p* < 0.05).

**Table 2 foods-14-01221-t002:** Color changes of cooked Antarctic krill treated at different hot-air drying temperatures.

Color	0	1	2	3	4
*L**	56.26 ± 6.08a	59.06 ± 6.20a	60.16 ± 4.53a	58.20 ± 3.51a	58.51 ± 3.61a
*a**	15.71 ± 2.50ab	16.63 ± 2.90a	17.00 ± 2.20ab	15.31 ± 2.10ab	14.45 ± 2.51b
*b**	27.07 ± 2.64a	26.85 ± 2.95a	29.33 ± 4.04a	25.39 ± 2.72a	22.32 ± 4.56b

Note: 1, 2, 3, and 4 present different hot-air drying temperatures at 80, 100, 120, and 150 °C, respectively. 0 means the control group that samples untreated by hot-air drying. Significant variations between samples are indicated by different letters (*p* < 0.05).

**Table 3 foods-14-01221-t003:** Analysis of volatile flavor compounds of cooked Antarctic krill treated at various hot-air drying temperatures by GC-MS (mg/kg).

	0	1	2	3	4
n-Hexane	25.46 ± 1.47a	31.75 ± 4.57a	27.50 ± 4.75a	40.70 ± 7.06a	45.96 ± 6.61a
Furan, tetrahydro-3-methyl	49.18 ± 18.62a	37.56 ± 29.85ab	19.19 ± 6.01b	14.12 ± 9.01b	15.65 ± 9.39b
Furan, tetrahydro-2-methyl	2.46 ± 2.11a	2.51 ± 2.78a	2.64 ± 3.50a	6.00 ± 5.32a	0.75 ± 0.12a
Methylamine, N, N-dimethyl	1.45 ± 0.61b	0.82 ± 0.11b	1.17 ± 0.27b	0.93 ± 0.92b	7.31 ± 3.76a
Ethanamine, N-methyl-	0.02 ± 0.02a	0.00 ± 0.00a	0.00 ± 0.00a	0.03 ± 0.02a	0.05 ± 0.04a
1-Butanol	0.04 ± 0.03a	0.01 ± 0.01b	0.01 ± 0.01b	0.02 ± 0.01ab	0.05 ± 0.02ab
3-Pentanone, 2-methyl-	0.03 ± 0.01ab	0.02 ± 0.01b	0.04 ± 0.01ab	0.18 ± 0.16a	0.18 ± 0.10a
(*E*)-2-Butenal	0.01 ± 0.00b	0.04 ± 0.00b	0.10 ± 0.03b	0.73 ± 0.55a	0.81 ± 0.13a
Cyclopentasiloxane, decamethyl-	0.03 ± 0.03a	0.07 ± 0.06a	0.06 ± 0.04a	0.03 ± 0.03a	0.00 ± 0.00a
2-Propen-1-ol	0.01 ± 0.01a	0.01 ± 0.01a	0.02 ± 0.01a	0.02 ± 0.02a	0.03 ± 0.03a
(*E*)-2-Pentenal	0.01 ± 0.00b	0.02 ± 0.01b	0.01 ± 0.00b	0.07 ± 0.04a	0.05 ± 0.02ab
2-Ethyl-trans-2-butenal	0.01 ± 0.01b	0.07 ± 0.03b	0.17 ± 0.05b	2.09 ± 2.09a	0.96 ± 0.74ab
Cyclopentanol	0.06 ± 0.06a	0.25 ± 0.25a	0.48 ± 0.47a	0.58 ± 0.08a	0.01 ± 0.01a
Heptanal	0.01 ± 0.01a	0.02 ± 0.01a	0.06 ± 0.02a	0.07 ± 0.06a	0.07 ± 0.07a
Dodecane	0.00 ± 0.00a	0.01 ± 0.01a	0.03 ± 0.03a	0.04 ± 0.06a	0.06 ± 0.06a
(*E*)-2-Hexenal	0.01 ± 0.00b	0.01 ± 0.01b	0.02 ± 0.01ab	0.02 ± 0.02ab	0.04 ± 0.01a
1,6-Cyclodecadiene	0.01 ± 0.01a	0.01 ± 0.01a	0.02 ± 0.01a	0.02 ± 0.02a	0.06 ± 0.06a
(*E*,*Z*)-3,6-Nonadien-1-ol	0.01 ± 0.01a	0.01 ± 0.01a	0.04 ± 0.01a	0.02 ± 0.02a	0.07 ± 0.09a
1-Pentanol	0.01 ± 0.00c	0.02 ± 0.00c	0.03 ± 0.00b	0.07 ± 0.01a	0.07 ± 0.01a
Styrene	0.05 ± 0.05a	0.06 ± 0.04a	0.06 ± 0.01a	0.04 ± 0.03a	0.04 ± 0.03a
Dicyclopropylmethanol, chlorodifluoroacetate	0.00 ± 0.00a	0.01 ± 0.01a	0.03 ± 0.01a	0.01 ± 0.01a	0.07 ± 0.09a
1-Penten-3-ol, 4-methyl-	0.01 ± 0.00b	0.02 ± 0.00ab	0.03 ± 0.00ab	0.04 ± 0.01a	0.03 ± 0.01ab
Cyclohexasiloxane, dodecamethyl-	0.03 ± 0.03a	0.10 ± 0.08a	0.10 ± 0.06a	0.07 ± 0.07a	0.03 ± 0.03a
(*Z*)-2-Penten-1-ol	0.05 ± 0.02c	0.11 ± 0.01bc	0.18 ± 0.02b	0.43 ± 0.10a	0.54 ± 0.08a
Pyrazine, 2,5-dimethyl-	0.00 ± 0.00b	0.01 ± 0.01b	0.02 ± 0.01b	0.03 ± 0.02b	0.10 ± 0.02a
1-Hepten-6-one, 2-methyl-	0.01 ± 0.01a	0.02 ± 0.01a	0.03 ± 0.01a	0.03 ± 0.03a	0.05 ± 0.05a
1,2-Butanediol	0.00 ± 0.00b	0.01 ± 0.00b	0.02 ± 0.00b	0.06 ± 0.03a	0.07 ± 0.03a
Cyclohexane, 1-ethyl-2-methyl-, trans-	0.00 ± 0.00b	0.00 ± 0.00b	0.01 ± 0.00ab	0.05 ± 0.05a	0.03 ± 0.02ab
Pyrazine, 2-ethyl-5-methyl-	0.00 ± 0.00a	0.00 ± 0.00a	0.01 ± 0.01a	0.02 ± 0.01a	0.03 ± 0.03a
Pyrazine, trimethyl-	0.00 ± 0.00b	0.00 ± 0.00b	0.00 ± 0.00b	0.01 ± 0.01b	0.10 ± 0.05a
alpha.-Hydroxyisobutyric acid, acetate	0.00 ± 0.00a	0.00 ± 0.00a	0.01 ± 0.00a	0.03 ± 0.03a	0.05 ± 0.07a
1-Octen-3-ol	0.01 ± 0.01a	0.03 ± 0.02a	0.04 ± 0.02a	0.07 ± 0.07a	0.06 ± 0.08a
2-Hexene, 3,5,5-trimethyl-	0.04 ± 0.04a	0.15 ± 0.09a	0.20 ± 0.11a	0.26 ± 0.26a	0.25 ± 0.16a
1-Hexanol, 2-ethyl-	0.06 ± 0.03ab	0.03 ± 0.03b	0.04 ± 0.03b	0.14 ± 0.08ab	0.22 ± 0.07a
(*E*,*E*)-2,4-Heptadienal	0.00 ± 0.00b	0.00 ± 0.00b	0.00 ± 0.00b	0.07 ± 0.05a	0.06 ± 0.05a
Isovanillin	0.02 ± 0.02a	0.05 ± 0.04a	0.06 ± 0.04a	0.04 ± 0.04a	0.03 ± 0.03a
Pyrrole	0.02 ± 0.00d	0.08 ± 0.02c	0.12 ± 0.01ab	0.13 ± 0.01bc	0.20 ± 0.03a
Benzaldehyde	0.00 ± 0.00b	0.02 ± 0.02b	0.01 ± 0.01b	0.05 ± 0.05ab	0.18 ± 0.12a
3,5-Octadien-2-one	0.01 ± 0.01b	0.01 ± 0.01b	0.00 ± 0.00b	0.17 ± 0.11a	0.30 ± 0.17a
Cyclohexene, 3-ethenyl-	0.02 ± 0.01c	0.03 ± 0.01c	0.05 ± 0.00c	0.11 ± 0.02b	0.21 ± 0.07c
Cyclohexanone, 4-methyl-	0.00 ± 0.00c	0.01 ± 0.00c	0.02 ± 0.00c	0.08 ± 0.01b	0.18 ± 0.05a
2(5H)-Furanone, 5-ethyl-	0.00 ± 0.00b	0.01 ± 0.01b	0.02 ± 0.00b	0.08 ± 0.01a	0.13 ± 0.07a
3-Dodecyne	0.02 ± 0.00b	0.02 ± 0.01b	0.03 ± 0.00b	0.05 ± 0.01b	0.09 ± 0.03a
2-Vinylfuran	0.01 ± 0.00d	0.02 ± 0.00a	0.02 ± 0.00ab	0.02 ± 0.00bc	0.02 ± 0.00c
2,4-Di-tert-butylphenol	0.03 ± 0.00a	0.03 ± 0.01a	0.03 ± 0.01a	0.03 ± 0.00a	0.03 ± 0.01a

Note: 1, 2, 3, and 4 present different hot-air drying temperatures at 80, 100, 120, and 150 °C, respectively. 0 means the control group that samples untreated by hot-air drying. Significant variations between samples are indicated by different letters (*p* < 0.05).

**Table 4 foods-14-01221-t004:** Fatty acid profiles of cooked Antarctic krill treated at different hot-air drying temperatures (g/100g).

	0	1	2	3	4
C4:0	0.15 ± 0.10a	0.24 ± 0.03a	0.01 ± 0.00a	0.11 ± 0.11a	0.00 ± 0.00a
C6:0	0.07 ± 0.02a	0.07 ± 0.01a	0.03 ± 0.02a	0.06 ± 0.03a	0.04 ± 0.04a
C8:0	0.03 ± 0.02a	0.02 ± 0.00a	0.02 ± 0.00a	0.01 ± 0.01a	0.02 ± 0.00a
C10:0	0.01 ± 0.00a	0.01 ± 0.00a	0.00 ± 0.00a	0.01 ± 0.00a	0.00 ± 0.00a
C12:0	0.06 ± 0.00a	0.06 ± 0.00a	0.05 ± 0.00a	0.06 ± 0.01a	0.05 ± 0.00a
C14:0	2.82 ± 0.14ab	3.06 ± 0.08ab	2.52 ± 0.16c	3.10 ± 0.29a	2.61 ± 0.16c
C14:1	0.04 ± 0.00a	0.04 ± 0.00a	0.04 ± 0.00a	0.08 ± 0.03a	0.04 ± 0.00a
C15:0	0.09 ± 0.00a	0.09 ± 0.00a	0.08 ± 0.00a	0.09 ± 0.03a	0.08 ± 0.00a
C16:0	5.23 ± 0.23a	5.28 ± 0.26a	5.11 ± 0.18a	3.93 ± 1.61a	5.04 ± 0.17a
C16:1	0.10 ± 0.00a	0.09 ± 0.00b	0.09 ± 0.00b	0.10 ± 0.02a	0.08 ± 0.00b
C17:0	0.41 ± 0.02a	0.40 ± 0.01a	0.44 ± 0.01a	0.30 ± 0.19a	0.39 ± 0.01a
C18:0	0.30 ± 0.02a	0.32 ± 0.01a	0.32 ± 0.01a	0.77 ± 0.60a	0.32 ± 0.01a
C18:1n9t	2.53 ± 0.15a	2.76 ± 0.04a	2.63 ± 0.09a	3.00 ± 0.65a	2.66 ± 0.07a
C18:2n6t	0.02 ± 0.00a	0.00 ± 0.00a	0.00 ± 0.00a	0.02 ± 0.00a	0.00 ± 0.00a
C18:2n6c	0.46 ± 0.03a	0.45 ± 0.01a	0.48 ± 0.02a	0.38 ± 0.13a	0.44 ± 0.01a
C18:3n3	0.17 ± 0.01a	0.15 ± 0.00a	0.15 ± 0.02a	0.11 ± 0.07a	0.13 ± 0.03a
C20:1	0.16 ± 0.01a	0.17 ± 0.00a	0.11 ± 0.02a	0.14 ± 0.07a	0.14 ± 0.02a
C20:5n3 (EPA)	3.05 ± 0.17a	3.17 ± 0.04a	3.34 ± 0.11a	3.43 ± 0.14a	3.14 ± 0.07a
C22:1n9	0.13 ± 0.00a	0.13 ± 0.00a	0.09 ± 0.01b	0.14 ± 0.02a	0.11 ± 0.01b
C20:4n6	0.05 ± 0.00a	0.06 ± 0.00a	0.00 ± 0.00a	0.06 ± 0.01a	0.03 ± 0.00a
C23:0	0.09 ± 0.00a	0.10 ± 0.00a	0.08 ± 0.01a	0.09 ± 0.00a	0.07 ± 0.01a
C24:0	0.07 ± 0.00a	0.05 ± 0.00a	0.00 ± 0.00a	0.00 ± 0.00a	0.00 ± 0.00a
C22:6n3 (DHA)	1.38 ± 0.11ab	1.33 ± 0.01ab	1.51 ± 0.02a	1.48 ± 0.24b	1.37 ± 0.01ab
EPA + DHA	4.43 ± 0.28	4.50 ± 0.05	4.85 ± 0.13	4.91 ± 0.38	4.51 ± 0.08

Note: 1, 2, 3, and 4 present different hot-air drying temperatures at 80, 100, 120, and 150 °C, respectively. 0 means the control group that samples untreated by hot-air drying. Significant variations between samples are indicated by different letters (*p* < 0.05).

## Data Availability

The raw data supporting the conclusions of this article will be made available by the authors upon request.

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
