# Peer review of "Effects of Hot-Air Drying Temperatures on Quality and Volatile Flavor Components of Cooked Antarctic krill (Euphausia superba)"

_foods, 2025, doi:10.3390/foods14071221_

Round 1
Reviewer 1 Report
Comments and Suggestions for Authors
The manuscript deals with the effects of 4 different hot air drying temperatures on the physicochemical properties and flavor of Antarctic krill. The experimental design is relatively reasonable, but the results and discussion do not adequately explain the flavor section. There are already a few studies that have investigated the effects of different drying temperatures and different drying techniques on quality and flavor, and therefore this work should be an extension of these studies. In this paper. a large number of volatile flavor components were detected, but it was not explained to which odors these compounds contribute, and furthermore, these results should be linked to a sensory evaluation. The sensory evaluation of the samples should also include taste as a key parameter. Based on this, the results of the determination of the volatile flavor components should be presented more clearly. PCA is a good choice, but it is quite weak to see and the fingerprint is completely unclear and not well interpreted. In general, the discussion is written very superficially without a deeper insight into the volatile flavor component results obtained. Furthermore, in Table 3 it should be marked between which components there are statistically significant differences in content depending on the drying temperature, which could serve to improve the discussion and draw conclusions.
Author Response
Dear editor,
we feel great thanks for your professional review work on our manuscript. According to your nice suggestions, we have mad extensive corrections to our previous manuscript, the detailed corrections marked in red are listed below.
The manuscript deals with the effects of 4 different hot air drying temperatures on the physicochemical properties and flavor of Antarctic krill. The experimental design is relatively reasonable, but the results and discussion do not adequately explain the flavor section. There are already a few studies that have investigated the effects of different drying temperatures and different drying techniques on quality and flavor, and therefore this work should be an extension of these studies. In this paper. a large number of volatile flavor components were detected, but it was not explained to which odors these compounds contribute, and furthermore, these results should be linked to a sensory evaluation. The sensory evaluation of the samples should also include taste as a key parameter. Based on this, the results of the determination of the volatile flavor components should be presented more clearly. PCA is a good choice, but it is quite weak to see and the fingerprint is completely unclear and not well interpreted. In general, the discussion is written very superficially without a deeper insight into the volatile flavor component results obtained. Furthermore, in Table 3 it should be marked between which components there are statistically significant differences in content depending on the drying temperature, which could serve to improve the discussion and draw conclusions.
A: Thanks for your comments, based on your comments, we have modified the following concerns 1) we have added the links between flavor and sensory evaluation in the manuscript, 2) we did not add the taste as a parameter in the sensory evaluation part because Antarctic krill has not been widely used as raw material for human, the concerns made us delete the taste as parameter for sensory evaluation, but we could put this parameter into the next publication; 3)we have added the new version of figures on PCA to make it much more clearer; 4) we have put more information on the discussion part in-depth; 5) we have marked the significant changes of components in table 3 in bold.
Thank you again for your valuable comments, hopefully our replies satisfy your questions.
Reviewer 2 Report
Comments and Suggestions for Authors
Dear Authors,
This interesting study is giving important information which could be applied in the practice during the processing of the Antarctic krill.
In the abstract are missing values of the most important results. Insert it.
Highlight in the introduction Antarctic krill significance in the human nutrition. Insert in manuscript.
What are the most important nutrients in the Antarctic krill? Insert in manuscript.
Highlight more about usage of krill oil. Insert in manuscript.
In the subsection 2.8 highlight that you conduct quantitative composition analysis.
In the table 1 insert drying temperatures.
When you discuss results from subsection 3.2. highlight that moisture content and water loss ratio analysis are important parameters since water is suitable medium for bacteria development and spoilage of food. Insert in manuscript.
Line 236 What is reason for the highest TBARS content in the group 1. Explain it.
In the subsection 3.6. highlight if some of quantified volatile flavor compounds is responsible for unwanted aroma and flavor of Antarctic krill. Insert it in manuscript.
Highlight in the discussion importance of balance between drying temperature which is important for prevention of spoilage and preservation of volatile compounds which contribute to sensory properties of krill.
In the conclusion do not repeat which you have already mentioned in the sections results and discussions. Give direct conclusion about findings in the manuscript.
Author Response
Dear Editor,
We appreciate your professional review work on our manuscript. Based on your suggestions, we made extensive corrections to our previous manuscript. The detailed corrections marked in red are listed below.
1.In the abstract are missing values of the most important results.
A: Thanks for your comments, we have added the most important results about the lipid oxidation and color changes of Antarctic krill during the hot-air drying treatment in the abstract part marked in lines 18-26.
2.Highlight in the introduction Antarctic krill significance in the human nutrition.
A:Thanks for your comments, we have add the significance of Antarctic krill for human "Relative studies have indicated that Antarctic krill compounds have the potential on anti-oxidation, anti-inflammation, anti-obesity and anti-diabetic benefits on human." in lines 42-44.
3. What are the most important nutrients in the Antarctic krill?
A: Thanks for your comments, based on the previous studies, the most important nutrients in Antarctic krill is the DHA, EPA and so on. since our studies worked on the effects of hot-air drying conditions on quality changes of Antarctic krill, it could be hard for us to demonstrate that issue. However, it is could be a good topic for us to analyze the nutrients changes of Antarctic krill for the future work.
4.Highlight more about usage of krill oil.
A:Thanks for your comments, Antarctic krill oil is critical for human use and it has been shown that many potential benefits on human, we have highlight the overall benefits of Antarctic krill products in the manuscript as "Relative studies have indicated that Antarctic krill compounds have the potential on anti-oxidation, anti-inflammation, anti-obesity and anti-diabetic benefits on human" in lines 42-44.
5.In the subsection 2.8 highlight that you conduct quantitative composition analysis.
A: Thanks for your comments, we have added a equation (2) to highlight how to calculate the fatty acid composition in lines 142-147.
6. In the table 1 insert drying temperatures.
A:Thanks for your comments, it is really good one. we have added the information "Note: 1, 2, 3, and 4 present different hot-air drying temperatures at 80, 100, 120, and 150 °C, respectively. 0 means the control group that samples untreated by hot-air drying" under all the tables.
7.When you discuss results from subsection 3.2. highlight that moisture content and water loss ratio analysis are important parameters since water is suitable medium for bacteria development and spoilage of food. Insert in manuscript.
A:Thanks for your comments, bacteria could not be detected under low water contents, we highlight this as "Our investigation found that cooked Antarctic krill remained WC values of 26.45 ± 0.03%, 26.15 ± 0.02%, 24.20 ± 0.01%, and 22.71 ± 0.01%. This water contents could efficiently inhibit the formation of bacteria in precooked-shrimp, thereby extending its shelf life and ensuring its quality [29,30]." in lines 369-373, however, it brings a new topic for the mould counting for the future work, we also added this part in lines 372-374.
8.Line 236 What is reason for the highest TBARS content in the group 1. Explain it.
A:Thanks for your comments, TBARS is a common marker used to measure lipid peroxidation, which indicates oxidative stress or damage to lipids. The highest TBARS content in Group 1 compared to other groups could be attributed to several factors. Some potential reasons for the observed difference. it could be the experimental conditions even though we have tried several times. To determine the exact cause, it would be necessary to analyze the specific experimental setup, treatments, and biological characteristics of the groups involved.
9.In the subsection 3.6. highlight if some of quantified volatile flavor compounds is responsible for unwanted aroma and flavor of Antarctic krill. Insert it in manuscript.
A:Thanks for your comments, we have described the unwanted aroma in the Antarctic krill in discussion part as "In the present study, a higher DMA value was observed. This chemical is an irritating gas with a fish-oil odor that is commonly found in aquatic products [32]. Oxidation products are useful indicators of the quality of aquatic products [33,34]. Therefore, proper DMA oxidation management is an effective method to maintain Antarctic krill quality." in lines 399-403.
10.Highlight in the discussion importance of balance between drying temperature which is important for prevention of spoilage and preservation of volatile compounds which contribute to sensory properties of krill.
A: Thanks for your comments, to find out the balance between drying temperature and other factors for maintain the quality of Antarctic krill is really hard. because the energy factor is the most critical one. therefore, we ends up to make a conclusion as "Considering the energy consumption of ships and the quality requirements of Antarctic krill, cooked Antarctic krill can be dried onboard by hot-air drying at temperatures of 100 or 120 °C." in lines 448-450. hopefully our reply satisfies your comments.
11.In the conclusion do not repeat which you have already mentioned in the sections results and discussions. Give direct conclusion about findings in the manuscript.
A:Thanks for your comments, we have modified the conclusion part.
Reviewer 3 Report
Comments and Suggestions for Authors
The study presents a relevant study on the effects of hot-air drying temperatures on the quality and volatile flavor components of cooked Antarctic krill (Euphausia superba). The research is well-structured, with an appropriate methodology, and presents important findings for optimizing the processing of this crustacean. However, some improvements can be implemented to enhance clarity, coherence, and the discussion of results. Below are section-specific comments with suggestions for improvement.
Abstract
The abstract could provide more detailed information about the methodology. Specific numerical data should be included. The significance of the obtained results should be more clearly emphasized.
Introduction
The study’s context is well established, but the justification for the investigation could be further emphasized. It is recommended to highlight the existing gap in the literature regarding the impact of drying temperatures on krill quality.
The introduction mentions some previous studies but could more clearly differentiate the present study from published works.
Some sentences are too long and could be rewritten for greater clarity.
Methodologies
The drying process could be described in more detail, including information on moisture control and temperature monitoring. Additionally, the authors could calculate the diffusivity of the drying process.
While the description of GC-MS in Section 2.10 is sufficient, a brief mention of column calibration conditions and the standards used could be added. Furthermore, provide the values for the Limit of Detection (LOD), Limit of Quantification (LOQ), and recovery percentage for the standards. Include the R² value of the standard.
Results and Discussion
The Results and Discussion section contains sufficient data for the article, but the results lack in-depth discussion. Explain the observed trends in your results in this section and provide more interpretation.
The sensory evaluation requires further discussion on how the different treatments affected the panelists' preferences.
The discussion on water distribution could be better connected to the impacts on product texture and stability.
The lipid oxidation section is well-supported but could further explore the implications of these findings for product preservation. The impact of different temperatures on fatty acid stability is a relevant point that could be further elaborated.
The analysis of volatile compounds is comprehensive, but the relationship between the identified compounds and sensory perception could be more detailed.
It is suggested to add a final paragraph connecting the findings with potential applications in the industry.
Conclusion
The Conclusions section should be more concise, including considerations on the significance of the results, the most novel findings, and areas for future research in this field.
Author Response
Dear Editor,
We appreciate your professional review work on our manuscript. Based on your suggestions, we made extensive corrections to our previous manuscript. The detailed corrections marked in red are listed below.
1. Abstract
The abstract could provide more detailed information about the methodology. Specific numerical data should be included, and the significance of the obtained results should be more clearly emphasized.
A: Thanks for your comments, we have added relative information of specific numerical data in abstract in lines 18-26.
2. Introduction
The study’s context is well established, but the justification for the investigation could be further emphasized. It is recommended to highlight the existing gap in the literature regarding the impact of drying temperatures on krill quality.
The introduction mentions some previous studies but could more clearly differentiate the present study from published works.
Some sentences are too long and could be rewritten for greater clarity.
A: Thanks for your comments, we have added some information and comparison about the our study to previous studies in the introduction part. At the same time, we also have shorted some sentences in this part.
3.Methodologies
The drying process could be described in more detail, including information on moisture control and temperature monitoring. Additionally, the authors could calculate the diffusivity of the drying process.
While the description of GC-MS in Section 2.10 is sufficient, a brief mention of column calibration conditions and the standards used could be added. Furthermore, provide the values for the Limit of Detection (LOD), Limit of Quantification (LOQ), and recovery percentage for the standards. Include the R² value of the standard.
A: Thanks for your comments, the water contents and water loss ratio were detected using the moisture analyzer. The machine could automatically analyze the water contents and water loss ratio, and the data recorded in the moisture analyzer. so we just setup the programs for weight of the samples, and the temperatures. we did not analyze the diffusivity as we cited the method described by "Zheng, Y.;Zhang, S.; Yang, L.; Wei, B. Guo, Q. Prevention of the Quality Degradation of Antarctic Krill (Euphausia superba) Meal through Two-Stage Drying. Foods 2024, 13(11), 1706. https://doi.org/10.3390/foods13111706." and "Fikry, M.; Benjakul, S.; Al-Ghamdi, S.; Tagrida, M.; Prodpran, T. Evaluating Kinetics of Convection Drying and Microstructure Characteristics of Asian Seabass Fish Skin without and with Ultrasound Pretreatment. Foods 2023, 12(16), 3024. https://doi.org/10.3390/foods12163024." it is a good topic that we could analyze the diffusive characteritsics in the future work.
For the GC-MS part, In the experiment, this was done for relative quantification, not external standard method quantification, and there is no standard curve. Additionally, this is headspace extraction injection, not solvent extraction, and does not involve indicators such as recovery rate.
4. Results and Discussion
The Results and Discussion section contains sufficient data for the article, but the results lack in-depth discussion. Explain the observed trends in your results in this section and provide more interpretation.
The sensory evaluation requires further discussion on how the different treatments affected the panelists' preferences.
The discussion on water distribution could be better connected to the impacts on product texture and stability.
The lipid oxidation section is well-supported but could further explore the implications of these findings for product preservation. The impact of different temperatures on fatty acid stability is a relevant point that could be further elaborated.
The analysis of volatile compounds is comprehensive, but the relationship between the identified compounds and sensory perception could be more detailed.
It is suggested to add a final paragraph connecting the findings with potential applications in the industry.
A: Thanks for your comments, we have modified our results and discussion part based on your comments, hopefully our reply satisfies your questions.
5.Conclusion
The Conclusions section should be more concise, including considerations on the significance of the results, the most novel findings, and areas for future research in this field.
A: Thanks for your comments, we have modified the conclusion part.
Round 2
Reviewer 1 Report
Comments and Suggestions for Authors
I am grateful that the comments were taken into account and the work was improved, and therefore I agree to publish the work in its present form.
Author Response
Dear Editor,
Thanks for your comments, we appreciate that the manuscript was improved by your valuable comments, and thanks that the manuscript could be accepted in this present form.
Thank you again.
Reviewer 2 Report
Comments and Suggestions for Authors
Dear Authors,
Thank you very much for revised version of your manuscript and answers on my questions and suggestions.
The answer which you have gave on suggestion under number 8 expalin important issues very well so you include it in the manuscript. It will expalin more problems which were investigated in your manuscript.
Wish you all the best in the future work,
Author Response
Dear Reviewer,
Thanks so much for your valuable comments, we have include the explanation related to number 8 in the manuscript in lines 262-267 based on your comments,
Thanks again.
Reviewer 3 Report
Comments and Suggestions for Authors
The study has been improved and can be accepted in its current form
Author Response
Dear Reviewer,
Thanks for your valuable comments which help the manuscript understandable. We appreciate that the manuscript could be accepted based on your comments.
Thanks again.